# Role of the Holoenzyme PP1-SPN in the Dephosphorylation of the RB Family of Tumor Suppressors During Cell Cycle

**DOI:** 10.3390/cancers13092226

**Published:** 2021-05-06

**Authors:** Eva M. Verdugo-Sivianes, Amancio Carnero

**Affiliations:** 1Instituto de Biomedicina de Sevilla, IBIS, Hospital Universitario Virgen del Rocio, Consejo Superior de Investigaciones Científicas, Universidad de Sevilla, Avda. Manuel Siurot s/n, 41013 Seville, Spain; everdugo-ibis@us.es; 2CIBERONC, Instituto de Salud Carlos III, 28029 Madrid, Spain

**Keywords:** RB family proteins, pocket proteins, cell cycle, phosphatase PP1, spinophilin, PPP1R9B, cancer, tumorigenesis

## Abstract

**Simple Summary:**

Cell cycle progression is highly regulated by modulating the phosphorylation status of retinoblastoma (RB) family proteins. This process is controlled by a balance in the action of kinases, such as the complexes formed by cyclin-dependent kinases (CDKs) and cyclins, and phosphatases, mainly the protein phosphatase 1 (PP1). However, while the phosphorylation of the RB family has been largely studied, its dephosphorylation is less known. Recently, the PP1-Spinophilin (SPN) holoenzyme has been described as the main phosphatase responsible for the dephosphorylation of RB proteins during the G0/G1 transition and at the end of G1. Here, we describe the regulation of the phosphorylation status of RB family proteins, giving importance not only to their inactivation by phosphorylation but also to their dephosphorylation to restore the cell cycle.

**Abstract:**

Cell cycle progression is highly regulated by modulating the phosphorylation status of the retinoblastoma protein (pRB) and the other two members of the RB family, p107 and p130. This process is controlled by a balance in the action of kinases, such as the complexes formed by cyclin-dependent kinases (CDKs) and cyclins, and phosphatases, mainly the protein phosphatase 1 (PP1). However, while the phosphorylation of the RB family has been largely studied, its dephosphorylation is less known. Phosphatases are holoenzymes formed by a catalytic subunit and a regulatory protein with substrate specificity. Recently, the PP1-Spinophilin (SPN) holoenzyme has been described as the main phosphatase responsible for the dephosphorylation of RB proteins during the G0/G1 transition and at the end of G1. Moreover, SPN has been described as a tumor suppressor dependent on PP1 in lung and breast tumors, where it promotes tumorigenesis by increasing the cancer stem cell pool. Therefore, a connection between the cell cycle and stem cell biology has also been proposed via SPN/PP1/RB proteins.

## 1. Introduction

During tumor development, cells undergo a series of genetic and/or epigenetic alterations, which gives them selective advantages over the environment, generating cancer cells. Many processes are involved in tumorigenesis, and one of the most important is the deregulation of the cell cycle. Therefore, cancer cells have undergone mutations deregulating the cell cycle that make them grow uncontrollably [1,2].

Cell cycle progression from one phase of the cycle to another is highly regulated through protein phosphorylation. In the G1 phase, there is a special checkpoint called restriction or R point at which the cell decides if it is ready to enter the cell cycle. The R point is controlled by the phosphorylation status of the retinoblastoma protein (pRB), a tumor suppressor protein whose main function is to inhibit cell cycle progression in G1 by binding E2F transcription factors and, thus, repressing E2F-target genes necessary to advance the cell cycle. Thus, until the pRB is phosphorylated and inhibited, cells cannot pass the R point and enter the cell cycle. Therefore, the G1/S transition is one of the most important checkpoints in the cell cycle [3,4,5,6,7,8]. Phosphorylation of the pRB inhibits its cell cycle restraining function by releasing E2F transcription factors. This phosphorylation is catalyzed by the complexes formed by cyclin-dependent kinases (CDKs) and cyclins. CDKs are serine-threonine kinases regulated by cyclins, proteins with cyclical expression whose levels increase and decrease drastically throughout the cell cycle, periodically activating CDKs [4,5,7,8,9,10,11,12,13,14,15,16,17]. The activity of CDKs/cyclins complexes can also be inhibited by the action of cyclin-dependent inhibitors, which can be divided into two families: the CIP/KIP family, composed of p21^CIP1^, p27^KIP1^ and p57^KIP2^, which inhibit most of CDKs/cyclin complexes; and the INK4 family, formed by p16^INK4A^, p15^INK4B^, p18^INK4C^ and p19^INK4D^, which inhibit G1 CDKs, especially CDK4 and CDK6 [4,5,8,15,17,18]. However, the dephosphorylation of the pRB to restore the cell cycle, which is mainly mediated by the protein phosphatase 1 (PP1), is also very important and much more overlooked. 

In this review, we describe the regulation of the phosphorylation status of the pRB and the other members of the RB family of tumor suppressors to emphasize not only their inactivation by phosphorylation but also their dephosphorylation to restore the cell cycle, two mechanisms that are frequently altered during tumorigenesis.

## 2. RB Family Proteins

The proteins of the pRB family or RB proteins are pRB itself, p107 (RBL1) and p130 (RBL2), three very similar proteins that share some biochemical properties and some functions [7,19,20,21]. These proteins are also known as pocket proteins because they have a domain *-the pocket-* capable of binding to different proteins and transcription factors. This pocket is composed of a smaller pocket, with two well-organized subdomains (A and B) separated by a less structured spacer region and the C-terminal region [4,17,18,22,23,24,25]. Specifically, the A/B subdomains bind to proteins containing the LXCXE motif, where “X” could be any amino acid. The PP1 phosphatase presents a variant of the motif (LXSXE) capable of binding to the pRB, while the E2F factors do not present this motif since their binding requires the entire pocket, including the C-terminal domain (Figure 1a) [4,7,17,18,22,26,27]. In addition, the pRB presents in this C-terminal region a special coupling site for E2F1 and a binding region for CDK2/cyclin A, CDK2/cyclin E, and PP1, while other CDKs bind to the N-terminus [7]. At least 16 phosphorylation residues are present in the pRB, all serine and threonine [15,17,28]. Phosphorylation of the pRB breaks the binding with different proteins; however, no kinase is capable of phosphorylating all of the pRB residues at the same time. The complete inactivation of the pRB requires sequential phosphorylation by different CDK/cyclin complexes, and depending on the residues that are phosphorylated, different proteins will dissociate sequentially, regulating the cycle-dependent genes differentially [4,18,26]. The structures of p107 and p130 are very similar to that of the pRB, but they are more related to each other (~50%) than to the pRB (20–30%). This is because their spacer region is larger and both present an insertion in the B subdomain of the small pocket and a region of homology at the N-terminus that allows them to act as inhibitors of CDKs (Figure 1a) [19,20,29,30,31,32,33]. Therefore, the pocket domain allows the association of RB proteins with many different proteins and transcription factors, some of them common and others specific to each protein so that, although there is not complete redundancy, there are some compensation mechanisms [20,21,23,27,34]. 

This family of proteins constitutes one of the major regulators of the cell cycle. They act by inhibiting transactivation mediated by activating E2F factors as well as forming complexes with E2F repressor factors to repress transcription and inhibit G1/S transition [35]. E2F levels vary throughout the cycle: while E2F1, E2F2, and E2F3 levels increase during the G1/S transition to induce proliferation, E2F4 and E2F5 are mostly expressed in resting cells [19]. However, the subset of E2F-dependent genes that each protein in the RB family regulates is different since each one interacts with different E2F factors: the pRB sequesters the activating factors E2F1-4, while p107 and p130 bind to repressor factors E2F4-5, although in the absence of E2F4, factors E2F1 and E2F3 can bind to p107 and p130 to compensate for their function [5,18,19,20,32,33,36,37]. E2F4 and E2F5 factors are expressed throughout the cycle, but during G0/G1, they bind p130 and p107 in the nucleus to form a repressor complex. In turn, the pRB sequesters E2F1, E2F2, and E2F3 to prevent binding to the corresponding promoters. At the end of G1, all the pocket proteins are phosphorylated and dissociate from the E2F factors so that E2F4-5 translocate to the cytoplasm and E2F1-3 bind to different promoters (Figure 1b) [35]. p130 (or p107) also mediates the repression of cell cycle genes as part of the DREAM (dimerization partner (DP), RB-like, E2F and multi-vulval class B (MuvB)) complex during quiescence [38]. Therefore, like the pRB, the activity of p107 and p130 is regulated during the cell cycle by controlling their state of phosphorylation in serine and threonine residues by the action of CDK/cyclin complexes in the middle/end of G1. When these proteins are dephosphorylated, they act as transcriptional repressors, while when they are phosphorylated, they are inactivated and dissociate from the E2Fs, allowing the transcription of genes involved in the cell cycle [19,20,21,35]. 

RB family proteins bind to numerous proteins, so they have other cellular functions beyond the control of the cell cycle. More than 200 proteins that interact with the pRB have been described, such as E2F transcription factors, cyclin D, MDM2, p53, and PP1, as well as other transcription factors and proteins related to differentiation, cell lineage identity, stemness invasion, apoptosis, senescence, angiogenesis, immune response and metabolism [4,17,18,22,23,28,40,41,42,43]. The pRB is an important tumor suppressor, and it is frequently inactivated, directly or indirectly, in many human tumors, promoting tumorigenesis [3,4,5,6,7,8,41,42,44]. Although p107 is hardly mutated in human tumors, and mice with mutations in p107 do not develop spontaneous tumors [30], the overexpression of hypophosphorylated p107 can induce G1 to stop in some cell types [32,36]. p130 is also not frequently mutated in cancer, but its levels are extremely low in some tumors due to its role in quiescence and differentiation [19]. Therefore, beyond genetic redundancy, the pRB could have some tumor suppressor functions that are not shared with p107 and p130 [5,7,20,23,34,39,42,45,46]. 

### Phosphorylation of Pocket Proteins 

One of the most important substrates of the CDKs/cyclins is the pRB [5,47]. The pRB is phosphorylated only on serine and threonine residues, and its phosphorylation state varies throughout the cycle. In addition, the pRB is synthesized de novo throughout the cycle. During G0 and at the beginning of G1, the pRB is hypophosphorylated and active, repressing the cell cycle through its interaction with E2F factors. Then, the pRB is progressively phosphorylated from the middle of G1 to M phase (mitosis) by different CDKs/cyclin complexes, becoming inactive and releasing E2F factors so that the genes involved in cell cycle progression can be expressed. Finally, the pRB is dephosphorylated by the PP1 phosphatase at the end of mitosis to return to its hypophosphorylated and active state during G0/G1 [3,4,5,6,8,9,10,12,14,15,17,18,20,21,25,35].

Complete inactivation of the pRB requires sequential phosphorylation by different CDKs/cyclin complexes. Initially, the pRB is phosphorylated by CDK4/6/cyclin D in the middle of G1, but this phosphorylation is partial and is not sufficient to release the E2F factors. For this to happen, the action of CDK2/cyclin E at the G1/S transition is necessary. However, only when the pRB has been phosphorylated by CDK4/cyclin D can it be phosphorylated by CDK2/cyclin E in the next phase so that cells cannot pass the R point if the pRB is not phosphorylated by both complexes. When the E2F transcription factors are released, certain cycle-dependent genes can be expressed, such as cyclin A, whose expression is delayed until the S phase. Phosphorylation of the pRB is maintained thanks to CDK2/cyclin A during the entry into S phase and until G2 when CDK1/cyclin A participates. Finally, CDK1/cyclin B phosphorylates and maintains this state during the mitosis phase (Figure 2) [4,5,6,7,10,12,13,14,15,16,17,18,20,21,47,48,49]. 

The pocket proteins collaborate in the control of the cell cycle since the pRB is expressed in both quiescent and proliferating cells, p130 is expressed mainly in quiescent cells, and p107 in proliferating cells [19,20,27,29,31,35]. The pRB levels hardly change throughout the cell cycle since its activity is controlled exclusively by the state of phosphorylation, but p107 and p130 fluctuate in the opposite way throughout the cell cycle (Figure 3a) [19,21,27,31,37,50]. 

All pocket proteins are apparently phosphorylated by CDK4/cyclin D, CDK2/cyclin E, and CDK2/cyclin A in the middle of the G1 to S phase [20,36,50,51,52]. p107 is hypophosphorylated and expressed at low levels when cells are differentiated or at rest during G0, but when cells are proliferating, p107 is phosphorylated and increases its levels from the middle of the G1 to S phase [30]. However, the phosphorylation state of p107 does not affect its levels [31]. p107 is mostly phosphorylated by CDK4/cyclin D, which inactivates it by dissociating the complex with E2F4 from the middle to the end of G1. CDK2/cyclin E and CDK2/cyclin A are also involved in the phosphorylation of p107 during the end of G1 and in the S phase, respectively, although they do not participate in its inactivation or in the dissociation of E2F4 [32,36]. Therefore, the p107/E2F4 complexes increase when leaving G0, then they dissociate and re-form at the end of G1 and especially during the S phase, in which the levels of p107 are very high [30,32]. In addition, although p107 is phosphorylated during G1, newly synthesized unphosphorylated p107 is also found in complex with E2F (Figure 3b) [32].

In the case of p130, three different phosphorylated forms (forms 1, 2, and 3), depending on their electrophoretic mobilities, have been described. Indeed, unlike the other RB family members, which are dephosphorylated in cell-cycle arrested cells, p130 is phosphorylated (forms 1 and 2) even in these cells [51,53]. When cells are at rest at the beginning of G1, the expression of p130 is very high, forming a complex with E2F4 to repress genes related to the cell cycle, mainly in their phosphorylated forms 1 and 2, which allow the stabilization and accumulation of p130 [19,21,27,31,37,50,51]. Phosphorylation of p130 begins at G0/G1 to reach its hyperphosphorylated form 3, which makes p130 a less stable protein to be degraded by the proteasome. Its levels fall drastically when cells are in the S phase. This degradation is induced by the phosphorylation of p130 by CDK4 in serine 672 [20,34,37,53,54]. When p130 is phosphorylated in the middle of G1, E2F4 is released, and E2F-dependent genes such as p107 and E2F1-2 are expressed, and the pRB/E2F1 and p107/E2F4 complexes are formed, the former being the complex more abundant in the middle and at the end of G1 (Figure 3b) [31,34]. Thus, as the G1 progresses, the levels of p130 decrease and those of p107 increase, which replaces p130, forming a complex with E2F4, mainly at the end of G1 and in the S phase [20,31,32,33,34]. Both E2F4 and the newly synthesized E2F1 activate the transcription of p107, so p107 is under the control of the pRB through an E2F-dependent transcriptional regulation, whereas p130 is not [20,31,55]. Furthermore, p130 needs to be phosphorylated by CDK4 and CDK2 to dissociate from E2F4, while p107 only needs CDK4 to break its bond with E2F4 [56]. 

On the other hand, when cells leave the cell cycle to enter quiescence, p130 accumulates, and p107 decreases, favoring the formation of the p130/E2F4 complexes and displacing both p107 and the pRB [20,31]. The accumulation of p130 could regulate the cell’s decision to exit the cell cycle before cells pass the R point [20,27]. However, during quiescence, the pRB is also complexed with E2F, so p130 and the pRB could have similar functions in this process [20,22,27,31,32,34,39]. Cell cycle progression by G1 must be considered differently depending on whether the cells come from G0, so there will be a large number of p130/E2F4 complexes or from a previous cycle, as there will be more pRB/E2F complexes [20,31,32,33,34]. Thus, the regulation mechanisms are different depending on the state of the cells, since when they leave quiescence, they must synthesize some of the cell cycle regulators [20,37].

Therefore, RB proteins cooperate to control the cell cycle in G0/G1, preventing the transcription of E2F-dependent genes [18,20,27,39,52]. Moreover, p107 and p130 repress other E2F-dependent genes than those repressed by the pRB [27,56], so although these proteins complement and compensate each other in certain cellular contexts, their function is not totally redundant [27]. 

## 3. Protein Phosphatase 1 (PP1)

Protein phosphorylation is the major mechanism for regulating cellular functions since more than 70% of eukaryotic proteins are regulated by phosphorylation, mainly at serine and threonine residues [57,58]. This process is controlled by a balance in the action of kinases and phosphatases; however, while there are more than 400 genes that code for serine-threonine kinases, there are fewer than 40 genes that code for serine-threonine phosphatases. This is because phosphatases are enzymes formed by a catalytic subunit capable of interacting with numerous regulatory proteins to form different complexes (holoenzymes) with different locations and substrate specificities [4,12,57,58]. The largest subfamily of serine-threonine phosphatases is the phosphoprotein phosphatases (PPP), responsible for 95% of the phosphatase activity in cells. The most studied are PP1, PP2A and PP2B, but one-third of the dephosphorylation events in eukaryotic cells are only carried out by PP1 [4,12,57,59,60,61,62,63,64,65]. 

PP1 is an enzyme involved in the regulation of many cellular processes, such as protein synthesis, transcription, apoptosis, and cell cycle progression [57,61,65,66]. In mammals, three genes (*PPP1CA*, *PPP1CB*, and *PPP1CC*) encode four isoforms of the catalytic subunit of PP1: PP1α, PP1β, PP1γ1, and PP1γ2. These four isoforms are expressed in all tissues and compartments, although PP1γ2 is only expressed in testes [10,57,58,64,67,68,69,70,71]. In addition, all isoforms are found in the nucleus, but PP1β and PP1γ present a special accumulation in the nucleolus [57]. The sequence of the three isoforms is highly conserved: 93% between PP1γ1 and PP1γ2 and 85% between PP1β and PP1γ2, although the N-terminal and the C-terminal present greater differences [10,57,58,64,67,68,69,70]. 

PP1 is involved in a large number of functions; however, it does not have substrate specificity by itself, and it needs to interact with multiple regulatory proteins. Thus, the catalytic subunit of PP1 (PPP1C) can bind and form different holoenzymes with multiple regulatory proteins (PPP1R), also known as regulatory interactors of protein phosphatase 1 (RIPPOs). RIPPOs direct PP1 towards specific substrates to perform specific functions, preventing the dephosphorylation of substrates by occupying the PP1 binding site or promoting the dephosphorylation by directing PP1 to specific cell locations. Additionally, some RIPPOs can be PP1 substrates [57,58,64,71,72,73,74,75]. Currently, approximately 200 regulatory proteins of PP1 are known, and most of them do not show similarity in their sequence [57,62,64,65,74]. Approximately 90% of PP1-interacting proteins bind to it through the PP1 binding motif RVxF, which generally consists of the consensus sequence: (K/R) (R/K) (V/I) (x) (F/W) where “x” can be any residue except F, I, M, Y, D or P [58,66,72,74]. However, this interaction is unique to each regulatory protein, so that mutations in the RVxF motif would prevent the binding of the regulatory protein to PP1 but would not affect the binding of the substrate or the formation of other holoenzymes [66]. In addition to the RVxF motif, there are other binding motifs that not only stabilize the binding to PP1 but also modulate its activity and specificity [57,58,64,72,73,74]. 

PP1 is the protein phosphatase responsible for dephosphorylating and activating the pRB from the exit of mitosis to the middle of G1 [3,12,18,39,67,76,77,78]. The interaction between the pRB and PP1 occurs through the PP1 binding motif, with a high affinity and is direct, without the need for regulatory proteins. However, these proteins serve as regulators of the pRB-specific PP1 activity and are context/tissue-dependent, since PP1 alone cannot regulate the pRB [21,39,70]. PP1 forms a complex with both the hypophosphorylated and hyperphosphorylated pRB; however, if the pRB is phosphorylated at certain residues, it cannot bind PP1, so PP1 needs the help of a regulatory protein to gain access to the pRB to dephosphorylate it. Indeed, the preferred sites for PP1 to dephosphorylate the pRB are T356 and S807/811. After dephosphorylation, PP1 remains bound to the hypophosphorylated pRB forming a complex that lasts until entry into G1, preventing its phosphorylation [4,14,67,76,77,79]. In addition, PP1 and the CDK/cyclin complexes share the binding site to the pRB, so there is a competition for the substrate between the kinase and the phosphatase activities [70]. On the other hand, p107 and p130 also present the PP1 binding motif. The interaction between these two proteins and PP1 has been described in double hybrid assays and by co-immunoprecipitation [70,80]. However, this interaction seems to be much weaker and lasts less time than that between the pRB and PP1 [21,50]. 

The regulation of PP1 is complex and is controlled by both phosphorylation by different kinases and the action of inhibitors and regulatory proteins that modulate its substrate specificity and activity [3,4,5,67,71,79,81,82]. On the one hand, PP1 is phosphorylated and inactivated during the cell cycle by CDK2/cyclin E, CDK2/cyclin A, CDK1/cyclin A and CDK1/cyclin B to avoid pRB dephosphorylation. Specifically, the phosphorylation of PP1 at the T320 residue plays an important role in the G1/S transition and during mitosis [3,71,83,84]. On the other hand, CDK1/cyclin B phosphorylates and inactivates PP1 with the help of the PP1 Inhibitor-2 (I2) during the onset and the middle of mitosis [4,85]. At the exit of mitosis, PP1 is activated by destroying the CDK1/cyclin B complex and by PP1 autodephosphorylation at residue T320 [14,18,57,76,83,85], which is inhibited during mitosis by the binding of Inhibitor-1 (I1) to PP1 [57,85]. 

The three isoforms of the catalytic subunit of PP1 bind the pRB similarly since the interaction region is conserved. In addition, all of them have the ability to dephosphorylate the pRB but present different activities in the different phases of the cycle [3,21,70,86]. During G1 and in the G1/S transition, PP1α is the main isoform that controls the pRB [61,79]. When cells enter mitosis, all isoforms are phosphorylated and inactivated since there is an increase in the phosphorylation of the PP1α protein at serine residues and PP1β and PP1γ1 proteins at threonine residues. Finally, at the end of the mitosis phase, PP1α and PP1β activity increases while PP1γ1 remains phosphorylated and with low activity [81,82]. Indeed, PP1β is the most active isoform during mitosis, but its activity does not persist during G1 [18,21,70,81,82,86]. Therefore, the dephosphorylation of the pRB is regulated in a sequential and temporal manner, and the three isoforms of the catalytic subunit of PP1 bind to different regulatory proteins to form different holoenzymes with different preferences for phosphorylation sites, similar to pRB phosphorylation by CDK/cyclin complexes [5,18,21,86]. In fact, different holoenzymes of PP1 may form during the cell cycle to control the dephosphorylation of the pRB since the three isoforms of the catalytic subunit of PP1 present different activities in the different phases of the cell cycle [3,4,5,67,79,81].

In this regard, the study of PP1 regulatory proteins involved in the cell cycle is essential since mutations in the catalytic subunit of PP1 or in the regulatory proteins that prevent binding to the pRB will promote phosphorylation of the pRB and, eventually, cell transformation [4,14,76,77,79]. 

## 4. SPN, a PP1 Regulatory Protein

Spinophilin (SPN), also known as PPP1R9B and NEURABIN-2, is a PP1 regulatory protein widely expressed in many tissues such as the brain, lung, testes, colon, breast, among others [87,88,89,90,91,92,93,94,95,96,97]. SPN presents in its structure a PP1 binding domain, but the SPN-PP1 interaction occurs not only through the RVxF motif but also by forming multiple interactions with different regions of PP1, including part of the C-terminus of PP1 [72,88,90,91,98,99]. Indeed, the structure of SPN suggests that it is a multifunctional protein that functions as a scaffold protein by recruiting many different proteins into different cell signaling pathways and promoting protein-protein interaction [88,90,91,98]. SPN has been shown to be located in the cytoplasm and in the plasma membrane of cells but could also be expressed in the nucleus [91,100]. We recently demonstrated that SPN co-localizes with PP1α and PP1γ in the nucleus and in the cytoplasm of cells [80].

One of the main functions of SPN is to help PP1 dephosphorylate the pRB [28,80,93]. SPN interacts with PP1α and PP1γ but not with PP1β [69,80]. In addition, SPN interacts specifically with both total and phosphorylated pRB (P-pRB) in Ser807/811, two of the preferred PP1 dephosphorylation sites [10,80]. SPN is also able to bind and dephosphorylate phosphorylated p107 (P-p107) in Ser975, a homologous residue to Ser807/811 in P-pRB, and phosphorylated p130 (P-p130) in Ser672, an important residue implicated in the stability of p130 during the cell cycle and a possible dephosphorylation site of PP1 [20,34,53,54,80]. Therefore, the PP1-SPN holoenzyme is not exclusive to the pRB but acts over all the pocket family proteins, although another phosphatase could dephosphorylate P-p107 and P-p130 in other contexts [80]. 

Cell cycle assays, in which cells were synchronized at G0 through serum deprivation or at the end of G1 after mimosine treatment, demonstrated that the PP1-SPN holoenzyme regulates the dephosphorylation of pocket proteins during the G0/G1 transition and at the end of G1. However, cell cycle assays in which cells were synchronized at the G2/M transition after nocodazole treatment showed that this holoenzyme does not act during mitosis when PP1β is the most active isoform and does not interact with SPN. Therefore, the PP1-SPN holoenzyme is formed by SPN and either PP1α or PP1γ and is involved in the dephosphorylation of pocket proteins exclusively during the G0/G1 transition and at the end of G1, but not during the G2/M transition or the mitosis phase [80]. Instead, PP1β could bind to a different PP1 regulatory protein at the exit of mitosis, but this other regulatory protein remains unidentified (Figure 4a) [4,14,67]. It has been reported that phosphatase nuclear targeting subunit (PNUTS) is a PP1 inhibitory protein with an important role in controlling PP1 activity during mitosis by inhibiting pRB dephosphorylation. However, PNUTS is only associated with a small proportion of PP1, and other proteins beyond PNUTS and SPN must regulate PP1 during the cell cycle [80,101]. PNUTS is a context-dependent PP1 regulatory protein, and the role of SPN in PP1 regulation and pocket protein dephosphorylation might also be dependent on the context, regarding either the cell cycle phase or the subcellular localization [80]. In addition, the pRB could function as a substrate or as a regulatory protein for PP1 since different subpopulations of the pRB perform different functions depending on the type of phosphorylation [77]. Different holoenzymes could be involved in the sequential control of pocket protein dephosphorylation during cell cycle progression, and each holoenzyme might have a distinct specificity for different phosphorylated residues, similar to CDK/cyclin complexes; therefore, initial dephosphorylation would be necessary to induce a conformational change before any other holoenzyme gains access to the different residues. In addition, whether the dephosphorylation of pocket proteins by PP1 in mitosis and in G1 occurs through a single mechanism or if different substrates are recognized by different holoenzymes must be determined [18,80].

On the other hand, SPN is phosphorylated by different protein kinases: protein kinase A (PKA) phosphorylates SPN in S97 and S177, calcium/calmodulin-dependent protein kinase II (CaMKII) phosphorylates SPN in S100 and S116, cyclin-dependent protein kinase-5 (CDK5) in S17 and mitogen-activated protein kinase-1 (MAPK1 or ERK2) in S15 and S205 [91,102,103,104], some of them also phosphorylate PP1. Indeed, phosphorylation of PP1 at Thr311/320 by CDK5 enhances its association with SPN, but phosphorylation of SPN by CDK5 at Ser17 is not responsible for the increased interaction between PP1 and SPN [105]. Although this enhanced interaction induced by CDK5 could add another layer of complexity to the regulation of RB proteins dephosphorylation during the cell cycle, it should be studied in depth to extract any conclusion.

In addition, SPN forms a different complex with PP1 and DCX, a microtubule-associated protein that binds to tubulin and actin. DCX is phosphorylated by CDK5, which prevents its binding to microtubules, and dephosphorylated by PP1 through interaction with SPN so that this axis regulates the maintenance of microtubules [106,107,108]. 

Therefore, the holoenzyme PP1-SPN performs different functions in the cell both in the nucleus and in the cytoplasm, depending on the association with different proteins.

## 5. SPN as a Tumor Suppressor Dependent on PP1 and pRB

The locus of *SPN* is located on chromosome 17 at the 17q21.33 position, a chromosomal region frequently associated with microsatellite instability and loss of heterozygosity and a high density of well-known tumor suppressor genes such as *BRCA1*. This loss of heterozygosity in the 17q21 region has been reported in different tumors, such as breast, ovarian, lung, prostate, colorectal, gastric, renal, and lung cancer [90,109,110,111,112,113,114,115]. Several studies suggested the existence of a new tumor suppressor gene located in the 17q21 region, and eventually, *SPN* was identified as this new gene [109]. Currently, *SPN* has been described as a tumor suppressor gene in the context of different human tumors, such as renal carcinomas, lung adenocarcinomas, ovarian carcinoma, chronic myeloid leukemia, gastric and colorectal cancer, head and neck carcinoma, hepatocellular carcinoma, and breast cancer [90,93,94,95,116,117]. 

Various studies in lung cancer have corroborated that *SPN* has a prognostic and predictive value in this type of tumor since the downregulation of this gene together with p53 mutations are associated with worse survival. In addition, a correlation between the decrease in SPN levels and low levels of the three catalytic subunits of PP1 was observed, and this combination was associated with a worse prognosis in squamous cell carcinoma. Loss of SPN has also shown a decrease in PP1 expression and activity in the brain tissue of SPN-knockout mice [118,119]. The SPN/PPP1C ratio could serve as a response biomarker due to its prognostic and predictive value in lung cancer. Indeed, a direct correlation was observed between the SPN/PPP1C ratio and the response to different drugs commonly used in the clinic, such as oxaliplatin and bortezomib; therefore, the SPN/PPP1C ratio could also be used as a therapy response marker in those types of tumors [96].

In breast cancer, SPN plays an important role as a tumor suppressor. In vivo studies using *Spn*-knockout mice reported that the absence of SPN decreased the life expectancy of mice and increased the number of spontaneous tumors such as lymphomas. *Spn*^−/−^ mice also presented an increased cell proliferation of certain tissues, such as the breast ducts, and both *Spn*^+/−^ and *Spn*^−/−^ mice showed more ramifications in this tissue. Indeed, *Spn*^−/−^ mice did not express SPN in the mammary ducts [92,120]. Mouse embryonic fibroblasts generated from *Spn*^−/−^ mice showed lower levels of PP1α and decreased PP1 activity, which in turn produced higher levels of phosphorylated pRB and increased p53 activity [89]. Additionally, the combination of the loss of SPN and p53 using p53-knockout mice induced preneoplastic lesions in the mammary glands, suggesting that the loss of SPN increases the p53 response similarly to oncogene-induced senescence. Thus, once spontaneous tumors appear, and p53 is lost, the loss of SPN increases their aggressiveness [90,92]. 

The loss of SPN has been reported in 15% of breast tumors, correlating with a higher histological grade, a less differentiated phenotype, and worse survival. In fact, both SPN and p53 are lost in triple-negative tumors, and this combination makes tumors more aggressive [121]. The downregulation of SPN in breast cancer cell lines increases some tumorigenic properties of the cells, such as the ability to proliferate or to form colonies, and some cancer stem cell properties, such as the formation of tumorspheres and the expression of stem cell genes. This effect depends on PP1 activity since the downregulation of PP1α mimics the effect of the downregulation of SPN [93,97]. Therefore, in tumor cells, the loss of SPN induces a proliferative response by reducing PP1α levels and increasing hyperphosphorylated and inactive pRB levels, which in turn activate p53 and neutralize the proliferative response. However, the loss or mutation of SPN is frequently associated with p53 mutations; therefore, in the absence of p53, the loss or mutation of SPN levels produces an increase in cell proliferation, and the tumorigenic properties of the cells are enhanced (Figure 4b) [80,89,90]. 

In addition, the downregulation of SPN also induces an increase in the stemness properties of the cells, such as the expression of some cancer stem cell markers (*NANOG, OCT4, SOX2,* and *KLF4*) and enrichment in CD44+/CD24- cells, cancer-initiating cells in breast tumors with stem cell properties [93,122]. Therefore, the loss of SPN in breast cancer induces an increase in the cancer stem cell pool, which worsens the response of those tumors to chemotherapy [80,93,97,123].

On the other hand, thirty-nine mutations in the region of interaction between SPN and PP1 have been identified [80]. The mutation of SPN, SPN-A566V, has an oncogenic effect since the expression of this mutation in breast cancer cell lines induces an increase in the tumorigenic and stemness properties of the cells depending on p53 mutations [80]. SPN-A566V affects the PP1 phosphatase activity of the holoenzyme, especially over the pocket proteins. Indeed, SPN-A566V did not interrupt the SPN-pRB interaction but decreased the capacity of the holoenzyme PP1-SPN to dephosphorylate P-pRB. The mutation of SPN affects the interaction between SPN and p107 and p130, and decrease the capacity of the holoenzyme PP1-SPN to dephosphorylate them [80]. Cells that overexpress SPN-A566V have high levels of the P-pRB, P-p107, and partially P-p130 during the G0/G1 transition and at the end of G1, which could mean that they have a shorter G1 phase in order to proliferate more rapidly. This mutation also induced an increase in the cancer stem cell pool and the expression of NANOG, OCT4, and SOX2 [80]. Recently, the pRB was reported to be directly involved in the transcriptional regulation of the pluripotency genes OCT4 and SOX2 [124]. When the pRB is dephosphorylated and active, the OCT4 and SOX2 promoters are inhibited [125]; thus, the P-pRB may promote OCT4/SOX2 expression in SPN-A566V cells, which in turn induces NANOG [126,127]. At the same time, OCT4 regulates the self-renewal and differentiation of embryonic stem cells and controls the cell cycle by increasing CDK/cyclin levels during the G1 phase and by preventing pRB dephosphorylation by PP1 [125,128,129]. Therefore, a connection between the cell cycle and stem cell biology was also proposed via SPN/PP1/pocket proteins, but further studies are needed to clarify whether the PP1-SPN holoenzyme plays any role in the OCT4/pRB self-regulatory circuit [80].

## 6. Conclusions

The pocket family of tumor suppressors constitutes one of the major regulators of the cell cycle through the interaction with E2F transcription factors. The regulation of the phosphorylation status of pocket proteins is controlled by a balance in the action of CDK/cyclins and different holoenzymes of PP1, like the recently characterized holoenzyme PP1-SPN, which is implicated in the dephosphorylation of RB proteins exclusively during the G0/G1 transition and at the end of G1. However, the dephosphorylation of pocket proteins is still not well understood, and further studies focusing on this mechanism are needed to improve the knowledge in this field. The development of specific inhibitors or activators that affect the interaction between PP1 and a regulatory protein and/or the regulation of a specific holoenzyme of PP1 could have great potential for cell cycle regulation and, consequently, for the treatment of numerous diseases such as cancer.

## Figures and Tables

**Figure 1 cancers-13-02226-f001:**
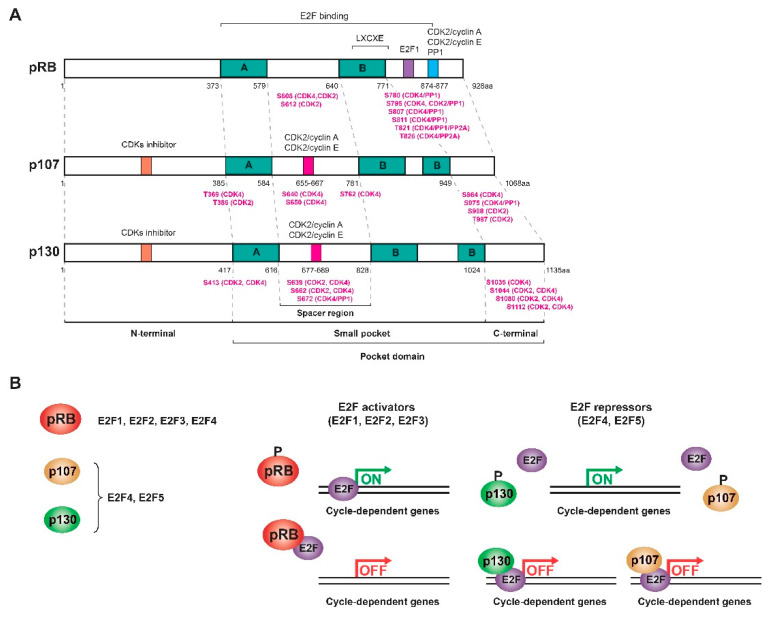
RB family proteins or pocket proteins. (**A**) Scheme of the structure of the three proteins of the RB family showing the different domains and motifs and the most relevant phosphorylated residues (highlighted in pink) with their respective kinases/phosphatases. (**B**) Interaction and regulation mechanisms of the pocket proteins with the different E2F transcription factors. Figure adapted from [19,20,21,39].

**Figure 2 cancers-13-02226-f002:**
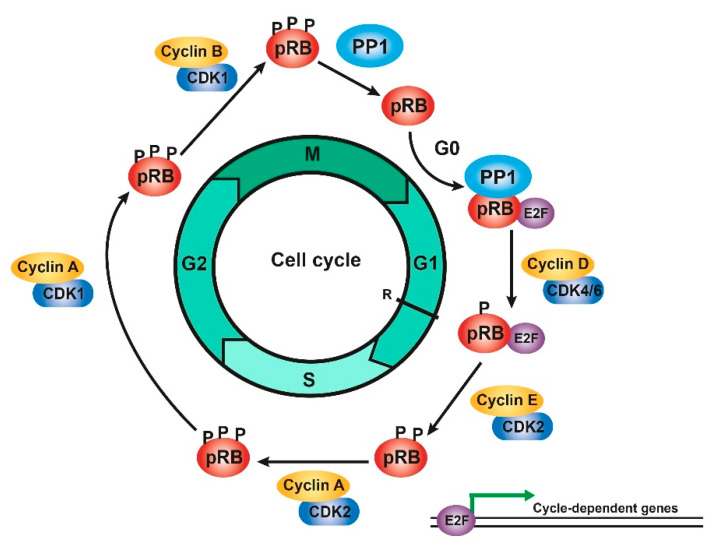
Phosphorylation/dephosphorylation of the pRB during cell cycle. Scheme of the regulation of the phosphorylation status of the pRB by different CDKs/cyclin and by PP1 during the different cell cycle phases. Figure adapted from [4,11,21].

**Figure 3 cancers-13-02226-f003:**
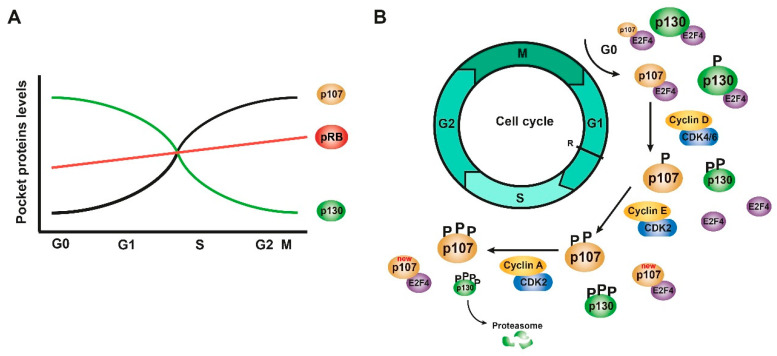
Pocket proteins collaborate in the control of the cell cycle. (**A**) Expression levels of the three proteins of the pocket family during the different phases of the cell cycle. (**B**) Scheme of the regulation of the phosphorylation and expression of p107 and p130 during G0, G1, and S phases. Figure adapted from [19,20,21,39].

**Figure 4 cancers-13-02226-f004:**
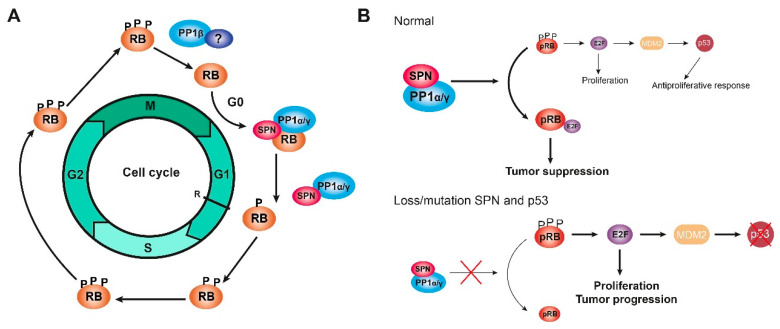
The holoenzyme PP1-SPN. (**A**) Scheme of the dephosphorylation of RB proteins (named here RB to refer to the three pocket proteins at the same time) during G1 by the holoenzyme PP1α/γ-SPN. The dephosphorylation of RB proteins might be regulated by PP1β in complex with an unknown regulatory protein during the end of mitosis. (**B**) Scheme of the mechanism of the holoenzyme PP1-SPN in tumorigenesis, comparing a normal situation (upper) and a tumoral situation in which SPN is lost/mutated, and p53 is mutated (bottom). Only when p53 is mutated, the loss/mutation of SPN can induce cell proliferation and tumor progression, evading the neutralizing response of p53. Figure adapted from [80,89,90].

## Data Availability

The study did not report any data.

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
