# Peer review of "Role of the Holoenzyme PP1-SPN in the Dephosphorylation of the RB Family of Tumor Suppressors During Cell Cycle"

_cancers, 2021, doi:10.3390/cancers13092226_

Round 1

Reviewer 1 Report

General comments:

  • Especially since it is an article submitted to the journal “Cancers”, I suggest that the authors clarify in the title of the manuscript what is the importance of this paper to the cancer research field
  • In the Introduction section, the authors should make a brief contextualization about the association between cell cycle and cancer
  • I consider that the organization of the topics 2 and 3 is not the better approach. I suggest to mention first the RB family proteins and the phosphorylation of these proteins, and then specify and detail the pRB, since this is the protein more explored in the next subtopics. For example:
  1. RB family proteins

     2.1. Phosphorylation of pocket proteins

     2.2. Retinoblastoma protein (pRB)

  • In the topic 4, in the sentence of the lines 178-182, the authors should clarify that some RIPPOs can be PP1 substrates
  • In the conclusion section, I think that the authors should explain what are the implications of these conclusions to cancer research, and contextualize their conclusions in cancer

Minor comments:

  • In the title, the authors should use the same approach for the name of two proteins (abbreviate or not): PP1/SPN or Protein phosphatase 1/Spinophilin
  • In the simple summary, line 16, substitute “responsible of” to “responsible for”
  • In the Introduction section, lines 44, 45 and 51, change “cycle” to “cell cycle”
  • In the caption of Figure 1, change “Phosphorylation” to “Phosphorylation/ dephosphorylation”
  • In the title of topic 4, change “Protein phosphatase PP1” to “Protein phosphatase 1 (PP1)”
  • In the topic 4, line 173, change “PP1ϒ1 y PP1ϒ2” to “PP1ϒ1 and PP1ϒ2” and “These three isoforms” to “These four isoforms”
  • In the topic 4, line 180, the authors should use the abbreviation RIPPOs instead of PPP1R. This abbreviation was proposed in the following manuscript: DOI: 10.1021/acscentsci.9b00909
  • In the topic 4, line 184, change “PP1 binding motif (RVxF)” to “PP1 binding motif RVxF”
  • In the topic 4, line 188, change “there are other interactions” to “there are other binding motifs”
  • In the topic 5, the second paragraph (lines 215-227), except the first sentence, should be moved for the previous topic
  • In the topic 6, line 268, since SPN does not appear in italics, the authors should substitute “this gene” for “this protein”. Alternatively, the authors can use SPN in italics
  • In the topic 6, line 279, change “Ppp1ca” to “PP1α”, since is mentioning the protein

Reviewer 2 Report

Cancers spinophilin/PP1 review

Overall, the review is well written, informative, and interesting. It discusses the role of dephosphorylation of major Rb proteins by PP1 and targeting of PP1 by one of its myriad targeting proteins, spinophilin. There are only a few areas that may be better described and/or mentioned in particular in regards to spinophilin. Specifically, additional comparing/contrasting spinophilin’s characterized roles in neurons and how roles in cell cycle progression may inform/be informed by roles in differentiated neurons, as this area has also been extensively studied for spinophilin. These are minor and may help inform the review in particular given that the title of the review is “the role of the holoenzyme PP1-SPINOPHILIN…” and so a little additional mention of spinophilin in the end sections may be warranted.

  1. Spinophilin expression levels are highest in the brain compared to other areas (Allen et al., PNAS 1997). While the authors state that spinophilin expression is ubiquitous and cite the Allen paper, they should mention that in this paper the author’s only observed limited expression of spinophilin in  tissues outside the brain; however, there was expression in the lung, testes, and adrenal glands, but no evaluation of breast tissue. I think it is not quite correct to say expression was “ubiquitous” as it may be somewhat restricted. More discussion of the low levels of expression in non-neuronal tissues should be given and the implications of this in spinophilin’s role as a tumor suppressor gene are important.
  2. PP1 has been shown to be phosphorylated by CDK5 (Li et al., JBC 2007, Hou et al., 2013 J. Cell Biol, ) (a non cyclin-dependent CDK). I am not aware if PP1 can be phosphorylated by the true CDKs mentioned in the review; however, it may be important to note that PP1 phosphorylation at Thr311/320 by CDK5 has been shown to previously enhance its association with spinophilin (Edler et al., ACS Chem Neuro 2018).
  3. Spinophilin is an actin-binding protein and is enriched in dendritic spines on neurons (Satoh et al., JBC 1998 and others). However, the authors are describing roles for spinophilin-targeted PP1 within the nucleus and/or nucleolus based on my reading of the review. Co-transfection of PP1 with spinophilin in heterologous cells leads to expression of both in regions co-localizing with actin and very little overlap with nuclear areas (although a nuclear stain was not performed; Carmody et al., 2008 FASEB J). A description or mention if spinophilin has ever been shown to be associated with F-actin in the nucleus would be worthwhile, or is a lack of spinophilin leading to increased expression of PP1 in the nucleus or at other intracellular organelles. This is an interesting point as CDK5 is known to phosphorylate spinophilin and regulate its interactions with actin (e.g. Futter et al., PNAS 2005).
  4. Carmody et al., 2008 FASEB J should probably be cited on line 212 as it suggests multiple PP1 binding domains on spinophilin that permit PP1 isoform specificity.
  5. Additional citations (Allen et al., Neurosciece 2006, Salek et al., J. Neurochem 2019) around line 268/269 may be warranted that show loss of spinophilin leads to decreased PP1 expression in brain tissue – suggesting a regulation of PP1 by spinophilin beyond just in cultured cells from KOs.

Additional minor grammatical issues

  1. In the summary and abstract, the statement “ Recently, the PP1-SPN holoenzyme has been described as the main responsible of the dephosphorylation of RB proteins during the G0/G1 transition 16 and at the end of G1.” Is missing a word after main. Maybe complex or repeat the word holoenzyme. Just read a little funny without that extra word.
  2. Line 125 extra word “the phosphorylation state of p107 does not affect at its levels”.
  3. Figure 3 has the word proteasoma. Should be proteasome or proteasomal?

Author Response

This manuscript is a resubmission of an earlier submission. The following is a list of the peer review reports and author responses from that submission.